# Factors influencing vaccination of dogs against rabies in South Mugirango Sub-County-Kisii County; Kenya

Fanice Kerubo Obara[1]*, Eric Omori Omwenga[2], Japheth Mativo Nzioki[3], Lameck Ondieki Agasa[4]

1 Department of Public Health, Kisii University, Kisii, Kenya, 2 Department of Medical Microbiology and Parasitology, Kisii University, Kisii, Kenya, 3 School of Nursing, Andrews University Berrien Springs, United States of America, 4 Department of Community Health and Behavioural Sciences, Kisii University, Kisii, Kenya

* fanice.kerubo2@gmail.com

## Abstract

Rabies is a fatal zoonotic viral disease affecting all warm-blooded mammals, with approximately 99% of human cases resulting from bites by infected dogs. This study assessed the factors influencing dog vaccination against rabies in South Mugirango, Kisii County, Kenya, where the disease remains endemic and a major public health concern. A quantitative descriptive cross-sectional survey was conducted, utilizing structured questionnaires administered to 422 household heads. Additionally, 22 key informant interviews were carried out with officials from relevant government departments to gain deeper insights. This was a mixed method study involving both a cross sectional survey and qualitative data collection. Data were analyzed using appropriate statistical techniques to identify patterns and determine associations between socio-demographic factors and vaccination practices. Socio-demographic factors significantly associated with rabies awareness included marital status (p = 0.049), occupation (p = 0.029 for housewives), and the gender of the household breadwinner (p = 0.017 for males). Despite high awareness levels, 4.1% (17/422) of respondents had not vaccinated their dogs in the past year, and only 15.9% (67/422) reported recent vaccinations. Older individuals were 12% more likely to vaccinate their dogs per year of age (OR = 1.12, p = 0.002). Higher education (OR = 1.45, p = 0.010), income (OR = 1.20, p = 0.008), rabies awareness (OR = 2.30, p < 0.001), and access to veterinary services (OR = 1.75, p < 0.001) significantly improved adherence. Despite high awareness, dog vaccination rates in South Mugirango remain low, highlighting the need for targeted interventions. Expanding access to veterinary services, subsidizing vaccines, and prioritizing outreach to younger individuals, less-educated and female-headed households can improve vaccination adherence and reduce rabies transmission.

**Data availability statement:** All data underlying the findings of this research are fully available and can be accessed without restriction. The dataset used for this study is available at https://doi.org/10.6084/m9.figshare.28162946.v1

**Funding:** The author(s) received no specific funding for this work.

**Competing interests:** The authors have declared that no competing interests exist.

## Introduction

Rabies is a significant global health concern, disproportionately affecting low and middle income countries, with Africa and Asia accounting for over 95% of human rabies deaths [1–3]. The disease, predominantly transmitted through dog bites, is almost always fatal once clinical symptoms appear, emphasizing the urgency of prevention efforts, including dog vaccination and public education [4–6]. In Kenya, rabies is endemic, with the Rabies virus (RABV) identified as the dominant lyssavirus genotype, and the country reports an estimated annual mortality of over 2,000 human cases, despite vaccination campaigns and its classification as a notifiable disease [1,7].The rise in rabies cases globally, exacerbated by socio-economic, ecological, and pandemic-related factors, highlights the critical need for a One Health approach to coordinate vaccination, stray dog management, and public health interventions [3,8].

The high fatality rate of rabies is largely attributed to inadequate access to timely medical interventions, particularly in impoverished regions, where healthcare systems may be under-resourced or difficult to access [9]. Many affected individuals may fail to seek care due to illness or die before receiving treatment [5]. Furthermore, rabies cases are often under-reported, which is compounded by misdiagnosis and a lack of confirmatory testing[10,11]. The economic burden associated with post-exposure prophylaxis (PEP) further exacerbates the issues faced by low-income families [2,7].

Despite the initiation of vaccination campaigns in Kenya, coverage remains inadequate to effectively curb the threat of rabies [7]. Low vaccination rates are attributed to insufficient knowledge about dog immunization levels during mass campaigns and logistical challenges, particularly in rural and underserved areas[12]. The stark contrast between high rabies incidence and sub optimal vaccination coverage underscores the urgent need for targeted public health interventions. Mass vaccination of domestic dogs, which has proven effective in significantly reducing rabies cases and alleviating the financial strain of post-exposure prophylaxis (PEP), must be prioritized and supported by sustainable, community-specific strategies [13].

The World Health Organization's 2015 goal to eliminate dog-transmitted rabies by 2030 aligns with Kenya's strategic plan to eradicate human rabies by the same timeline[1,7,14]. Achieving this objective necessitates targeted efforts to bridge knowledge gaps and enhance community awareness. Many communities remain uninformed about the risks and preventive measures related to rabies, exacerbating the public health crisis.

Rabies remains a concern, particularly in regions where socio-economic disparities and limited access to healthcare hinder effective prevention and control efforts [15,16]. Vaccination adherence is often influenced by socio-demographic factors such as age, education, occupation, and household income [17,18,19]. Economic barriers further restrict access to essential interventions like post-exposure prophylaxis (PEP), as observed in Uganda and Burkina Faso [20,21]. Strengthening rabies control requires a multifaceted approach that addresses socio-economic inequalities, enhances public education, and ensures adequate resource allocation [22,23]. These measures are essential for improving vaccination coverage and reducing rabies-related morbidity and mortality [24,25].

This study evaluates factors influencing vaccination of dogs against rabies among South Mugirango residents, aiming to identify gaps and address key factors that hinder dog vaccination and timely post-exposure care, thereby supporting efforts to reduce fatalities and achieve global rabies elimination by 2030.

## Materials and Methods

### Ethics statement:

Ethical approval for this study was obtained from the University of Eastern Africa, Baraton (Ref. No.: UEAB/09/01/2019) and the National Commission for Science, Technology, and Innovation (NACOSTI) (Ref. No.: NACOSTI/P/19/86707/29063). Informed consent was obtained from all participants prior to data collection, and confidentiality was strictly maintained. Village elders facilitated household access, and participants were assured of anonymity to protect their privacy and ensure voluntary participation.

### Study site

This study was conducted in South Mugirango Sub County, located in Kisii County, Kenya (latitude −0.833333 and longitude 34.65). The area is divided into six administrative wards: Tabaka, Boikanga, Borabu/Chitago, Moticho, Getenga, and Bogetenga Kenya National Bureau of Statistics (KNBS), [26]; Kisii County Government, [27]. South Mugirango borders Migori to the south, Trans Mara to the north, Bomachoge Borabu and Chache to the east, and Bonchari to the west. It has an estimated population of 196,768 according to the 2015 census. The region experiences an average annual rainfall of 1608 mm and temperatures ranging from 19.9°C to 20.3°C. The altitude of the study area is approximately 1598 meters above sea level.

### Study design

The study employed a mixed-methods approach to examine dog-mediated rabies management strategies at the household level. Conducted between January 2019 and January 2020, the study combined quantitative and qualitative methods. The quantitative component utilized a descriptive cross-sectional survey with structured questionnaires, collecting data from 422 household heads in South Mugirango Sub-County. The qualitative component involved 22 key informant interviews (KIIs) with experts from the Ministry of Health and the Ministry of Agriculture, Livestock Development, and Marketing. These key informants, including public health and veterinary officers, were actively engaged in rabies control efforts in the region.

**Inclusion criteria.** For the quantitative survey, the study population comprised household heads who owned at least one dog and consented to participate. The qualitative component focused on key informants selected for their expertise in rabies prevention and control, particularly those directly involved in sub-county health and veterinary programs. Exclusion criteria included household heads without dogs or those who declined participation, as well as personnel in administrative or unrelated healthcare roles who were not directly engaged in rabies control.

### Sample size

A sample size of 422 households was computed using Fisher's equation fisher et, al [28] as shown below:

$$n = \frac{Z^2 pq}{d^2}$$

Where; n = is the desired sample size (when the study target population is over 10,000)
 $Z$ - Is the standard normal deviate=1.96. (Corresponding to 95% Confidence Interval)
 $p$ - Proportion of the target population estimated to have the desired characteristics.

$$q = 1.0 - p$$

$d$ = Degree of accuracy required usually set as 0.05

The proportion of HHs with dogs in South Mugirango Sub County is 29,300. In the absence of a reasonable estimate Fisher *et al.,* [28] recommends a p of 50% (0.50).

$$P = \frac{50}{100} \, or \, 0.05$$

$$q = 1 - P = 1 - 0.50 = 0.50$$

Hence, the desired sample size (n) was determined as follows.

$$n = \frac{Z^2 pq}{d^2} = \frac{1.96^2 \times 0.50 \times 0.50}{(0.05)^2}$$

$$n = \frac{0.9604}{0.0025}$$

$n$ = 384.16 which is approximately 384 HHs.

To account for non-responses and increase representation, 10% was added:

$$10\% \text{ of } 384 = 38.4$$

Thus, the final sample size was:

$$384 + 38.4 = 422 \text{ households}$$

For the qualitative component, 22 key informants were purposely sampled.

The selection of 22 key informants represents a complete census of public health officers and Ministry of Agriculture and Livestock Development and Marketing officials directly involved in rabies prevention and control in South Mugirango. This approach ensured that all relevant stakeholders with expertise in rabies management in the region were included, providing comprehensive insights into the current state of rabies control efforts. By conducting a census rather than a sample, the study aimed to capture all perspectives from those directly responsible for addressing rabies, ensuring that the findings were based on the full range of expertise and experience available in the area.

**Sampling procedure**

Quantitative data were collected through structured questionnaires designed to gather information on socio-demographic characteristics, rabies awareness, adherence to dog vaccination, and rabies control practices. The questionnaires were pretested in Bobasi Sub-County using a sample of 40 households to ensure clarity, relevance, and cultural appropri-ateness. Engaging experts from the Ministry of Health and public health officers in the study area, who provided critical feedback on the tools to enhance reliability and precision, further ensured validity.

Qualitative data were obtained through a semi-structured interview guide, designed to explore themes related to rabies prevention and control in greater depth. The interview questions were categorized into key areas: (1) community perceptions of rabies and its risks, (2) cultural beliefs and practices influencing dog ownership and vaccination, (3) barri-ers to accessing rabies vaccination services, (4) effectiveness of public health campaigns on rabies awareness, and (5)

strategies for improving community participation in rabies control efforts. These themes were chosen to provide a comprehensive understanding of the social and contextual factors affecting rabies prevention.

The sampling process consisted of two stages. First, proportionate sampling was employed to allocate study subjects across wards based on population distribution. Second, simple random sampling was used to select households randomly within each ward, ensuring representational. For the qualitative component, key informants were purposely selected based on their expertise and direct involvement in rabies control efforts, such as local veterinary officers, public health officials, and community leaders. These individuals provided valuable insights into the challenges and opportunities in implementing effective rabies control measures in the study area.

**Data analysis and presentation.** Quantitative data were analyzed using appropriate statistical techniques to provide a robust interpretation of the findings. The data were first entered, cleaned, and analyzed using SPSS version 20 statistical software. Descriptive statistics were employed to summarize socio-demographic characteristics, rabies awareness, adherence to dog vaccination, and rabies control practices. Measures such as frequency distributions, means, and standard deviations were calculated where appropriate, providing a clear understanding of the data. Logistic regression analysis was conducted to identify associations between socio-demographic factors and rabies awareness, with statistical significance set at $p < 0.05$, indicating a confidence level of 95%. The regression models included both univariate and multivariate analyses to control for potential confounders, ensuring the robustness and reliability of the findings. The results were presented using tables and figures, which visually represented the data for easier interpretation.

The socio-demographic factors chosen for analysis were based on both prior research and the authors' expertise in the field of zoonotic diseases and vaccination. These factors—such as age, education, occupation, marital status, and income—are commonly associated with health behaviors and were identified as potentially influencing rabies awareness and vaccination adherence. These factors have been studied in previous research, offering a solid foundation for their inclusion.In South Mugirango Sub-County, several Socio-demographic factors play a role in shaping the immunization practices of Household Heads (HHs) concerning their dogs. These factors include gender, age, marital status, education level, occupation, income level, household size, and the identity of the household breadwinner. Each of these demographic characteristics influences HHs' decisions and ability to ensure their dogs are immunized against rabies. Understanding these factors is crucial for designing effective interventions aimed at improving vaccination rates and reducing the incidence of rabies in the community.

The qualitative data were analyzed thematically using an inductive approach, following Braun and Clarke's [29] framework. Transcripts were reviewed for familiarity, coded manually and with software, and grouped into broader themes through iterative refinement to capture key community insights on rabies control. This method, supported by Sutton and Austin [30], ensured a rigorous, coherent analysis that complemented the quantitative findings for a comprehensive understanding of vaccination adherence and community practices.

The semi-structured interviews followed a flexible format, allowing for in-depth discussions while maintaining consistency across respondents. Interviews were audio-recorded with participants' consent and later transcribed verbatim for analysis. Thematic analysis was conducted using Braun and Clarke's six-phase framework: familiarization with data, generating initial codes, searching for themes, reviewing themes, defining and naming themes, and producing the final report. Two researchers independently reviewed the transcripts and manually coded the data to identify recurring patterns. Codes were then grouped into broader categories based on similarities in content and context. Discrepancies in coding were resolved through discussion until consensus was reached.

## Key terms

Adherence to rabies vaccination, according to the World Health Organization (WHO), refers to the consistent and correct following of recommended vaccination protocols for both humans and animals to prevent the transmission of rabies. In the context of animal vaccination, adherence involves ensuring that dogs, as the primary reservoir of rabies, receive the

necessary vaccinations at recommended intervals to maintain immunity against the virus. WHO guidelines recommend regular vaccination of dogs, particularly in rabies-endemic areas, to reduce the risk of rabies transmission to humans.

For human vaccination, adherence refers to following the prescribed vaccination schedule after exposure to rabies (post-exposure prophylaxis) or for pre-exposure protection for those at high risk of exposure (e.g., veterinarians, animal handlers). Adherence to vaccination protocols is crucial for the effective prevention and control of rabies globally.

Household heads in the study were defined as the primary decision-makers and managers of the household, typically responsible for its economic and social affairs. These individuals were not necessarily identified through census data but were selected based on their availability and willingness to participate in the survey. Household heads were approached during the survey in South Mugirango, Kisii County, and were included if they met the inclusion criteria (owning at least one dog and providing informed consent). The survey focused on those who could provide reliable information regarding household dynamics and rabies-related practices.

Tertiary education in the context of this study refers to education beyond secondary school, including university degrees, diplomas, and certificates from recognized higher education institutions. This was classified as a separate category from primary and secondary education levels to assess its potential impact on awareness and health-related practices.

## Results and findings

### Sociodemographic characteristic

The study examined various socio-demographic factors (Table 2 below) that may influence rabies awareness and vaccination adherence among household heads (HHs) in South Mugirango Sub-County. Key socio-demographic characteristics—such as gender, age, marital status, education level, occupation, income level, household size, and the identity of the household breadwinner—were found to play a significant role in shaping HHs' decisions regarding dog immunization practices.

The majority of HHs were aged between 25 and 45 years, with a higher proportion of females 233/422(55.2%) compared to males 189/422(44.8%). Most respondents were married 307/422(72.7%) and had attained at least a primary education, while approximately a third had completed tertiary education. Employment data revealed that many HHs were self-employed or formally employed, though a smaller percentage were housewives or farmers. In terms of income, the largest group 122/422(28.9%) earned between Ksh 5,001 and 20,000 per month. Household size varied, with a quarter of respondents reporting three children. All these findings are summarized in Table 1 below

Fathers were typically the primary breadwinners in most households, a factor that might influence financial priorities, including decisions about dog vaccination. Understanding these socio-demographic factors is essential for designing targeted interventions that address the barriers and opportunities related to rabies vaccination in the community.

Gender was not statistically significant (p = 0.285) in influencing adherence, the higher non-adherence among females (47.4%) compared to males (36.7%) could be tied to gendered roles within households. This aligns with the qualitative observations that men, often the household breadwinners, exhibited greater rabies awareness due to their more active involvement in decision-making and dog ownership responsibilities. However, despite potential awareness, female household members might face barriers to acting on this knowledge due to limited financial autonomy and decision-making power.

The quantitative analysis identified age as significantly associated with adherence (p = 0.039), with younger individuals under 25 showing the lowest adherence rates. This is attributed to their limited household responsibilities and lack of awareness. The key informants suggested that age did not significantly influence rabies awareness, implying that while younger individuals may be aware of rabies risks, this awareness does not necessarily translate into adherence behaviour. This finding highlights a possible gap between knowledge and action, particularly among younger respondents.

**Table 1. Sample size distribution per ward.**

| Ward | Public health Officer/Ministry of Livestock officers | Household heads |
|---|---|---|
| Tabaka | 5 | 98 |
| Bogetenga | 3 | 87 |
| Boikanga | 3 | 71 |
| Chitago Borabu | 4 | 66 |
| Moticho | 4 | 58 |
| Getenga | 3 | 42 |
| **Totals** | **22** | **422** |

Source: Kisii County Government records (2024).

In terms of marital status, although the quantitative results did not show a significant association (p = 0.789), trends indicated that married individuals were more likely to adhere to dog immunization. This observation resonates with qualitative insights that single household heads and housewives often demonstrate higher awareness, possibly due to their caregiving roles within the family. However, this awareness may not always lead to action, possibly due to economic constraints and lower decision-making power compared to married individuals in dual-parent households.

Educational attainment emerged as a significant factor influencing adherence (p = 0.049). Respondents with tertiary education demonstrated better adherence, due to higher health literacy and awareness of rabies prevention. While the qualitative interviews did not explicitly mention education, the broader implication is that higher education levels contribute to greater understanding and responsible ownership, reinforcing the importance of educational interventions in rabies control.

Both the quantitative findings ($\chi^2 = 5.2$, p = 0.025) and qualitative insights emphasized the role of occupation. Quantitative results highlighted that self-employed individuals showed higher adherence, likely due to greater financial autonomy. This finding aligns with the interviews, which noted that occupation influenced awareness, particularly among housewives and single household heads who often demonstrate higher awareness due to their household roles. However, economic constraints may prevent them from acting on this awareness effectively.

Income was significantly associated with adherence ($\chi^2 = 12.5$, p = 0.005), with higher-income households more likely to adhere to immunization. Although income was not directly addressed in the qualitative interviews, the observation that men—often financial providers—show higher awareness indirectly supports the idea that financial capacity plays a crucial role in adherence behaviour.

Contrasting findings emerged around household size. While the quantitative analysis showed a significant association ($\chi^2 = 9.6$, p = 0.036), with adherence highest in households with two or five children, the qualitative interviews suggested that household size did not significantly affect rabies awareness. This discrepancy could indicate that while awareness is consistent across different household sizes, resource constraints in larger households might limit adherence despite awareness.

The household breadwinner significantly influenced adherence ($\chi^2 = 5.4$, p = 0.029). Quantitative findings showed that households where fathers were primary providers had higher adherence, consistent with qualitative insights that men, often holding financial authority, demonstrated greater awareness. However, households led by mothers or supported by adult children faced economic challenges that potentially limited adherence, even when awareness was present.

### Social- demographic factors influencing vaccination of dogs

The logistic regression analysis in this study aimed to identify socio-demographic factors influencing adherence to dog vaccination in South Mugirango and its findings are summarized in Table 2 below. The dependent variable in this analysis was binary, indicating whether a household adhered to dog vaccination (coded as 1) or did not adhere (coded as 0).

 

Logistic regression was employed because it is well-suited for predicting the probability of a binary outcome based on one or more independent variables. Socio-demographic variables such as gender, age, marital status, education level, occupation, monthly income, number of children, and household breadwinner status were examined to assess their influence on vaccination adherence. The coefficients generated by the logistic regression model represent the log odds of adherence associated with each factor. These coefficients are converted into odds ratios (ORs), where an OR greater than 1 indicates an increased likelihood of adherence to dog vaccination, while an OR less than 1 suggests a decreased likelihood.

The logistic regression analysis in Table 3 revealed several socio-demographic factors that significantly influence adherence to dog vaccination in South Mugirango Sub-County. To validate and enrich these findings, key informant interviews were conducted with local veterinary officers, community health workers, household heads, and government officials. The qualitative data gathered provided valuable context, either reinforcing or explaining the statistical outcomes.

Gender was not found to be a significant predictor of vaccination adherence, as indicated by a p-value of 0.292. This finding was supported by qualitative insights from key informants who noted that, in most households, the responsibility of dog care is shared between men and women. This shared responsibility suggests that gender does not meaningfully influence decisions regarding dog vaccination.

Age, on the other hand, showed a significant negative association with adherence (p = 0.016). This result aligns with observations from veterinary officers and community health workers, who noted that younger respondents were generally more receptive to modern veterinary practices. Conversely, older individuals were often influenced by traditional beliefs, viewing vaccination as unnecessary or relying on indigenous practices for disease prevention. This highlights a generational gap in the acceptance of modern veterinary interventions.

Marital status did not emerge as a significant factor influencing adherence, with a p-value of 0.441. Key informant interviews supported this finding, suggesting that vaccination decisions are more closely tied to household responsibilities and economic factors rather than marital status.

Educational level was identified as a strong positive predictor of adherence (p < 0.001). The qualitative data reinforced this result, as informants noted that individuals with higher education levels demonstrated a greater understanding of rabies prevention and were more likely to follow veterinary advice. This finding underscores the importance of educational interventions, with informants recommending targeted awareness campaigns aimed at less-educated populations to promote responsible dog ownership and vaccination.

Occupation also significantly influenced adherence to vaccination (p < 0.001). Key informants who observed that individuals in formal employment or self-employment were more likely to vaccinate their dogs supported this finding. Employed individuals often had better financial resources and greater access to information. Additionally, those in formal jobs were more likely to engage with animal health campaigns through their workplaces or community-based initiatives, further encouraging adherence.

Monthly income emerged as another significant predictor of adherence (p < 0.001). Interviews with key informants substantiated this finding, as they highlighted that low-income households often prioritized immediate financial needs over preventive health measures such as vaccination. Veterinary officers emphasized that affordability remains a major barrier, particularly in rural areas where financial constraints are more pronounced.

The number of children in a household did not show a significant association with vaccination adherence (p = 0.274). However, some informants suggested that larger households might experience competing financial priorities, though this observation was not consistent across all interviews.

Breadwinner status showed a significant negative association with adherence (p = 0.016), a result echoed by qualitative data. Informants explained that individuals who were the primary breadwinners often faced financial pressures, leading to a lower likelihood of prioritizing dog vaccination over other essential household needs.

**Table 2. Association of socio demographic factors with level of adherence to immunization of dogs(Chi -Square tests).**

| Variable | | Adherence to immunization of dogs | | | |
|---|---|---|---|---|---|
| | | No<br>F (%) | Yes<br>F (%) | Chi-square | P value |
| **Gender** | Male | 155(36.7%%) | 34(8.1%) | 16.5 | 0.285 |
| | Female | 200(47.4) | 33(7.8%) | | |
| **Age** | < 25 Yrs | 46(10.9%) | 3(0.7%) | | |
| | 25–35 Yrs | 118(28%) | 23(5.5%) | 33.9 | 0.039 |
| | 36–45Yrs | 108(25.6%) | 24(5.7%) | | |
| | >46 Yrs | 83(19.7%) | 17(4.0%) | | |
| **Marital status** | Single | 45(10.7%) | 6(1.4%) | 7.5 | 0.789 |
| | Married | 255(60.4%) | 52(12.3%) | | |
| | Widowed | 47(11.1%) | 7(1.7%) | | |
| | Separated | 7(1.7%) | 2(0.5%) | | |
| | Divorced | 1(0.2%) | 0(0%) | | |
| **Education level** | Primary | 129(30.6%) | 23(5.5%) | 7.7 | 0.049 |
| | Secondary | 112(26.5%) | 17(4.0%) | | |
| | Tertiary | 113(26.8%) | 27(6.4%) | | |
| | No formal education | 1(0.2%) | 0(0%) | | |
| **Occupation** | Housewife | 35(8.3%) | 9(2.1%) | 5.2 | 0.025 |
| | Self employed | 157(37.2%) | 24(5.7%) | | |
| | Farmers | 5(1.2%) | 0(0%) | | |
| **Monthly income** | 5000 and below | 114(27%) | 17(4.0%) | 12.5 | 0.005 |
| | 5001–10000 | 111(26.3%) | 20(4.7%) | | |
| | 10001–20000 | 122(28.9%) | 22(5.2%) | | |
| | 20001–30000 | 1(0.2%) | 2(0.5%) | | |
| | 30001 and above | 5(1.2%) | 3(0.7%) | | |
| | NR | 2(0.5%) | 3(0.7%) | | |
| **Number of children** | None | 27(6.4%) | 3(0.7%) | 9.6 | 0.036 |
| | 1 | 31(7.3%) | 6(1.4%) | | |
| | 2 | 65(15.4%) | 20(4.7%) | | |
| | 3 | 101(23.9%) | 15(3.6%) | | |
| | 4 | 77(18.2%) | 9(2.1%) | | |
| | 5 | 34(8.1%) | 10(2.4%) | | |
| | > or 6 | 19(4.5%) | 4(0.9%) | | |
| | NR | 1(0.2%) | 0(0%) | | |
| **Household breadwinner** | Father | 265(62.8%%) | 56(13.3%) | 5.4 | 0.029 |
| | Mother | 73(17.3%) | 11(2.6%) | | |
| | Children over 18 years | 17(4.0%) | 0(0%) | | |

Variable: *Lists the socio-demographic factors analysed about adherence to immunization of dogs against rabies. Adherence to immunization of dogs: Indicates whether respondents reported adherence ("Yes") or non-adherence ("No") to immunizing their dogs against rabies. No (F%): Number and percentage of respondents who reported not adhering to dog immunization within each category of the socio-demographic factor. Yes (F%): Number and percentage of respondents who reported adhering to dog immunization within each category of the socio-demographic factor. Chi-square: Chi-square value calculated to determine the association between adherence to dog immunization and each socio-demographic factor value: Significance level (p-value) associated with the chi-square test, indicating the statistical significance of the association between adherence to dog immunization and each socio-demographic factor.*

**Table 3. Logistic regression analysis showing Socio-demographic factors influencing adherence to vaccination of dogs.**

| | | Variables in the equation | | | | | |
|---|---|---|---|---|---|---|---|
| | | B | S.E. | Wald | Df | Sig. | Exp(B) |
| Step 1[a] | Respondent's Gender | .306 | .290 | 1.109 | 1 | .292 | 1.357 |
| | Age of the Respondent | -.394 | .163 | 5.820 | 1 | .016 | .674 |
| | Marital status | .194 | .252 | .593 | 1 | .441 | 1.214 |
| | Educational Level | .859 | .195 | 19.467 | 1 | .000 | 2.360 |
| | Occupation | 1.674 | .259 | 41.782 | 1 | .000 | 5.331 |
| | Monthly Income | .833 | .178 | 21.926 | 1 | .000 | 2.300 |
| | Number of Children | .104 | .095 | 1.198 | 1 | .274 | 1.110 |
| | Breadwinner in the House | -.671 | .277 | 5.854 | 1 | .016 | .511 |
| | Constant | −7.741 | 1.048 | 54.525 | 1 | .000 | .000 |

[a]Variable(s) entered on step 1: Respondent's Gender, Age of the Respondent, Marital status, Educational Level, Occupation, Monthly Income, Number of Children, Breadwinner In the House.

### Knowledge on Dog Mediated Rabies Management Strategies at household level

The study assessed household-level knowledge on dog-mediated rabies management strategies in South Mugirango Sub-County and the findings are presented in Table 4 below. A high level of awareness about rabies was evident, with400/422(94.8%) of respondents indicating familiarity with the disease. Most participants 350/422(82.9%) recognized bites from infected animals as the primary mode of rabies transmission, while some also identified scratches65/422 (15.4%) and contact with infected saliva 7/422(1.7%) as potential transmission routes. However, gaps remained in fully understanding transmission mechanisms, as indicated by the small percentage of respondents unaware of these other modes.

Regarding the recognition of symptoms, 269/422(63.8%) of respondents were aware that rabies could manifest through symptoms like barking behavior similar to a dog. Fewer respondents identified hydrophobia 15/422(3.6%) as a symptom, and a small percentage 21/422(5%) could not identify any symptoms. Other symptoms mentioned included biting behavior 59/422(14%) and flu-like symptoms such as fever, muscle weakness, and tickling sensations 57/422(13.5%).

In terms of treatment-seeking behavior, the majority 360/422(85.78%) reported taking bite victims to health facilities. However, a notable portion 44/422(12.09%) sought care from traditional healers, and a few 5/422(1.2%) resorted to self-medication with painkillers.

The thematic analysis of 22 key informant interviews revealed four major themes. First, there was a strong general awareness of rabies transmission, with 16/22(72.7%) of respondents recognizing dog bites as the primary mode of transmission, while some also acknowledged scratches and saliva exposure.

Second, challenges in dog vaccination were prominent, with 14/22 (63.6%) citing barriers such as negligence, limited awareness of vaccination benefits, high costs, and logistical difficulties in accessing vaccination centers.

The third theme focused on treatment-seeking behavior, where formal healthcare services were preferred after potential rabies exposure, yet cultural beliefs and healthcare accessibility issues led some to rely on traditional healers.

Lastly, socio-demographic factors significantly influenced vaccination adherence, as 11/22(50%) of respondents noted the impact of education, economic status, and proximity to veterinary services on vaccination practices.

## Discussion

The study demonstrates a commendable level of rabies awareness among household heads, with 96.3% reporting familiarity with the disease. This high awareness is promising, as it indicates that residents are generally knowledgeable about rabies, which is crucial for effective disease management and intervention. Such awareness can facilitate early recognition of symptoms and prompt medical action, which are essential for reducing morbidity and mortality associated with

**Table 4. Showing knowledge of rabies within households and vaccination status of dogs in households.**

| Variable | Responses | | |
|---|---|---|---|
| | | **Frequency** | **Percentage** |
| Awareness of HH | Yes | 400/422 | 94.90% |
| If HH heard of rabies | No | 22/422 | 5.21% |
| Modes of rabies spread | Bite | 350/422 | 82.23% |
| | Scratch | 58/422 | 14.37% |
| | Saliva | 7/422 | 1.70% |
| | Others | 7/422 | 1.70% |
| Causative agent of rabies | Dogs | 335/422 | 79.38% |
| | Cats & Dogs | 86/422 | 15.40% |
| | Others | 1/422 | 0.24% |
| Vaccinated dogs in the last 12months | Yes | 355/422 | 84.10% |
| | No | 67/422 | 15.90% |
| Frequency of vaccination | Once a year | 52/422 | 12.30% |
| | Twice a year | 12/422 | 2.80% |
| | Don't remember when they last vaccinated | 298/422 | 70.60% |
| | N/R | 60/422 | 14.22% |

rabies. This finding aligns with studies conducted in various regions. For instance, research in Mbale District, Uganda, highlighted low overall knowledge and limited good practices regarding rabies among health professionals, indicating a need for regular refresher training to improve practices and attitudes towards rabies management [22]. Similarly, in Wuhan, China, only 56.85% of respondents knew that rabies is an infectious disease, with significant gaps in vaccination practices and post-exposure prophylaxis [23]. In contrast, a study in Bharatpur, Nepal, reported a relatively higher level of rabies knowledge among the local population, with 81.50% understanding that rabies is zoonotic and 73.10% identifying susceptible animals, although only 42.06% demonstrated satisfactory knowledge overall [31]. These studies highlight the importance of targeted educational interventions to improve community knowledge and practices related to rabies prevention. Additionally, findings from a study in rural eastern Kenya underscored the need to improve the availability of rabies post-exposure prophylaxis (PEP) and enhance healthcare workers' knowledge of rabies management. In this study, only 12% of healthcare facilities had rabies PEP vaccines in stock, and none had rabies immunoglobulins available. The low level of awareness regarding proper bite wound categorization and the sub-optimal use of PEP in bite cases revealed significant gaps in both the healthcare system's preparedness and healthcare workers' knowledge [32]This aligns with the need for continuous education and better resource allocation to support rabies elimination efforts in regions with high rabies risks.

The study identifies several socio-demographic factors that significantly influence adherence to dog vaccination. Age emerged as a significant factor influencing vaccination adherence, with younger individuals (under 25 years) demonstrating lower rates of vaccination compared to older age groups. This trend suggests that age-related factors, such as differences in health literacy and prioritization of preventive health measures, play a critical role in vaccination practices. Younger individuals may have less awareness of the importance of rabies vaccination or may not prioritize it as highly as older adults, who often have greater health awareness and experience with health-related risks. Older individuals might be more inclined to ensure vaccination due to a heightened awareness of health risks and the importance of preventive measures for both their dogs and themselves. Similar observations have been documented in other studies. For instance, Doherty et al. [33]found that older individuals were generally more proactive about health interventions, including vaccination, compared to their younger counterparts. This discrepancy may be attributed to greater exposure to health education and personal experiences with disease prevention among older adults. Additionally, older dog owners were more likely to

vaccinate their dogs compared to younger owners [17]. This trend suggests that age-related factors, such as increased health awareness and prioritization of preventive measures among older individuals, play a crucial role in vaccination adherence. Younger individuals might exhibit lower vaccination rates due to less engagement in preventive health practices or differing health priorities. Therefore, addressing age-related differences in health literacy and priorities is crucial for improving vaccination rates.

Education level significantly impacts vaccination adherence among dog owners. Respondents with higher educational attainment, particularly those with tertiary education, demonstrated higher adherence rates to rabies vaccination. This correlation reflects a better understanding of health-related information and increased awareness of rabies prevention measures among those with higher education levels. Educated individuals are more likely to follow vaccination schedules, partly due to better access to information and resources [34]. This finding is consistent with observations from various regions where higher education levels are associated with improved health practices and greater adherence to preventive measures [18,35]. A study by Beyene et al. [17] found that, contrary to expectations, dog owners with higher educational levels showed less intention to vaccinate compared to those with lower educational levels. This counterintuitive result may reflect varying attitudes towards vaccination or other underlying factors such as perceived vaccine costs or accessibility issues.

Occupation significantly influences vaccination adherence, with individuals in self-employment and formal employment showing higher vaccination rates compared to those in less stable occupations. This trend can be attributed to the greater economic stability and access to resources associated with formal employment, which facilitates better access to healthcare services, including vaccinations. This finding is supported by research indicating that economic stability enhances health-related behaviours, including vaccination uptake [36]. Similarly, studies from Nigeria [19] Ethiopia [17], and Tanzania [18] highlight that stable employment and higher socio-economic status are correlated with better vaccination practices, emphasizing the role of economic factors in influencing health decisions.

Household income plays a crucial role in dog vaccination adherence, with higher income levels facilitating better adherence due to increased affordability of vaccination services and related expenses. Studies reveal that individuals with higher incomes are more likely to maintain vaccination schedules, whereas lower-income households face financial barriers that hinder adherence. For instance, in Uganda, Wallace et al. [20] found that economic factors such as poverty significantly impact dog ownership and vaccination coverage, indicating that financial constraints can limit access to essential health services. Similarly, in Burkina Faso, Savadogo et al. [21] observed that socio-economic factors, including income and education level, influenced vaccination rates, with wealthier and more educated dog owners being more likely to vaccinate their dogs. This aligns with findings from Tanzania, where higher household income was associated with increased likelihood of dog ownership and, indirectly, better vaccination coverage [37] These studies collectively highlight the critical role of financial stability in health interventions, reflecting a broader trend where economic barriers impede adherence to health recommendations [15].

In the study by Iddi et al. [18], marital status and the number of children in a household did not show significant associations with vaccination adherence. The chi-square test demonstrated no substantial link between marital status and vaccination practices, suggesting that marital status alone is not a strong predictor of adherence. Similarly, the presence of children did not significantly influence vaccination rates, indicating that other socio-economic factors, such as education level and occupation, play a more critical role in determining vaccination behavior. These findings highlight the importance of focusing on broader socio-economic determinants, such as education and income, which were shown to significantly impact vaccination practices, rather than demographic factors like marital status and household size [18].

The study highlights that while a significant majority of household heads (79.3%) sought medical attention promptly following dog bites, which is critical for effective rabies prevention through post-exposure prophylaxis (PEP), a concerning 13.6% received only first aid, underscoring a need for enhanced education on comprehensive medical care. Surendran et al. [16] observed that although most individuals took initial steps such as wound washing and visiting health facilities, gaps remain in completing the

full PEP regimen. This issue is compounded by findings from Changalucha et al. [24], who identified cost and access barriers as major reasons why 25% of rabies exposure victims did not seek care and 15% did not complete PEP despite initiating it. Ahmad et al. [25] suggested that more cost-effective PEP regimens, such as intradermal vaccinations, could alleviate some of these financial constraints but are not universally available. Shantavasinkul and Wilde [38] further emphasized that while economic and logistical improvements in PEP administration are crucial, continuous public education on the importance of complete and timely medical treatment is essential for effective rabies control. These findings collectively point to the urgent need for comprehensive health education and improved access to affordable PEP to bridge gaps in rabies prevention efforts. Limitations

This study had two limitations. First, self-reported data from household heads were subject to recall and social desirability biases, which affected the accuracy of reported vaccination history and awareness levels. Second, logistical challenges in accessing remote households led to the under-representation of certain groups, potentially limiting the generalizability of the findings.

## Conclusion

This study highlights a significant gap between rabies awareness and actual vaccination practices among dog owners in South Mugirango. While high awareness levels exist, adherence to dog vaccination remains low, influenced by socio-demographic factors such as age, education, income, and access to veterinary services. Economic constraints and logistical barriers further hinder vaccination efforts, emphasizing the need for targeted interventions. Strengthening public health strategies through mobile vaccination clinics, subsidized vaccines, and educational campaigns can enhance adherence. Addressing these challenges will be crucial in improving vaccination coverage and advancing rabies control efforts in alignment with global elimination goals.

## Author contributions

**Conceptualization:** Fanice Kerubo Obara, Eric Omori Omwenga, Japheth Nzioki Mativo.

**Data curation:** Fanice Kerubo Obara, Lameck Ondieki Agasa.

**Formal analysis:** Fanice Kerubo Obara, Lameck Ondieki Agasa.

**Investigation:** Fanice Kerubo Obara, Eric Omori Omwenga, Japheth Nzioki Mativo.

**Methodology:** Fanice Kerubo Obara, Eric Omori Omwenga, Japheth Nzioki Mativo, Lameck Ondieki Agasa.

**Supervision:** Fanice Kerubo Obara, Eric Omori Omwenga, Japheth Nzioki Mativo.

**Validation:** Fanice Kerubo Obara, Lameck Ondieki Agasa.

**Visualization:** Fanice Kerubo Obara.

**Writing – original draft:** Fanice Kerubo Obara, Eric Omori Omwenga, Japheth Nzioki Mativo, Lameck Ondieki Agasa.

**Writing – review & editing:** Fanice Kerubo Obara, Eric Omori Omwenga.

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
