## [Decision Letter · Decision Letter 0]

19 Nov 2024

PGPH-D-24-02047

Factors Influencing Vaccination of Dogs against Rabies within Household Heads in South Mugirango Sub-County-Kisii County; Kenya

Dear Dr. Fanice Kerubo Obara,

Thank you for submitting your manuscript to PLOS Global Public Health. After careful consideration, we feel that it has merit but does not fully meet PLOS Global Public Health’s publication criteria as it currently stands. Therefore, we invite you to submit a revised version of the manuscript that addresses the points raised during the review process.

We look forward to receiving your revised manuscript.

Kind regards,

Sukanta Chowdury, Ph.D

Academic Editor

Reviewers' comments:

**Reviewer #1** : The manuscript, "Factors Influencing Vaccination of Dogs against Rabies within Household Heads in South Mugirango Sub-County-Kisii County; Kenya," investigates the awareness and vaccination practices related to rabies among household heads in South Mugirango, Kenya. The study employs a mixed-methods approach, combining a descriptive cross-sectional survey with 422 household heads and key informant interviews with 22 health officials. Key findings indicate a high level of rabies awareness (94.8%) among respondents, with significant socio-demographic factors influencing vaccination adherence, including age, education level, occupation, and income. However, despite the high awareness, vaccination rates were low, with only 15.9% of respondents reporting recent vaccinations of their dogs. The study highlights a discrepancy between awareness and practice, attributing low vaccination rates to logistical challenges, inadequate funding, and socio-economic barriers. The authors emphasize the need for targeted public health interventions to enhance educational efforts about rabies prevention, improve vaccine access, and address economic barriers. While the manuscript provides valuable insights into rabies management and vaccination adherence in a specific Kenyan context, it could benefit from a more detailed exploration of study limitations and a stronger emphasis on actionable recommendations for public health strategies.

The study could benefit from broader contextualization by linking its findings to existing public health frameworks and rabies control programs in Kenya and comparing its results to previous interventions and their success rates to understand the correlation between awareness and vaccination uptake. Potential biases may arise in participant selection and self-reported vaccination behaviors, mainly due to social desirability bias common in survey studies. Additionally, the disconnect between high awareness and low vaccination adherence necessitates further exploration, as awareness alone does not ensure changes in health behaviors; interventions should address underlying beliefs, socio-economic barriers, and logistical challenges. Future research should focus on developing intervention strategies that engage the community and provide education tailored to demographic groups less likely to vaccinate their dogs, as literature indicates that targeted outreach can effectively bridge existing knowledge gaps.

I have a few more observations that the authors could consider. The line number should be inserted in the revised manuscript so that it is easy for the reviewer to make comments.

1. The introduction is lengthy, with repeats, and very generic. It must be shortened and better focused on the specific study. Consider following the logical flow of the scientific introduction. Stating the significance of addressing the specific gaps identified in previous studies would enhance the justification for the research.

2. The sample size is based on a well-defined statistical formula, which is commendable. The calculation and inclusion of a non-response rate are straightforward. However, details on how the structured questionnaires were validated or pretested should be included to ensure the reliability and validity of the data collection tools.

3. It is noted that structured questionnaires were employed alongside key informant interviews. However, a more detailed description of the questions, categories, or themes covered in the qualitative aspects would enhance transparency.

4. Data Analysis: The use of SPSS for data analysis is standard practice, although describing specific techniques (e.g., types of tests) applied would clarify the analytical approach. The qualitative data analysis process needs more elaboration. Details on coding, thematic analysis, or how themes were identified should be included.

5. Results: Avoid repeating data in the text already in the table except for the key findings. The manuscript effectively presents results through tables, aiding in comprehension. Key statistics are included, although some sections could benefit from visual representation (e.g., graphs showing vaccination rates over time). The results are relevant and significant to the research question but could be organized more logically. Distinguishing between awareness and direct vaccination practice results could clarify the impact of awareness on behaviors.

6. The limitations of the study are not explicitly discussed.

7. Consider providing a more operationalized set of recommendations based on findings.

**Reviewer #2:**  General comments: The study aims to fill an important gap of discerning the factors associated with rabies awareness and vaccination. The study does a good job of developing a design and sampling protocol. However, the analyses do not hold up to the study design. The analyses as they stand do not fulfill the aims described by the author. Different levels of factors need to be tested separately for statistically significant findings. The materials and methods do not adequately explain how the study was conducted making it hard to reproduce and reuse. This limits the impact of the study on contribution to the public health awareness of the communities being studied for rabies exposure. The writing style is not consistent and harder to follow. Based on this, I would recommend that the authors revisit all their data and re-analyze according to the aims proposed in the study. The authors have clearly collected enough data to redo most of the statistical work. Major restructuring and addition of supplementary material is necessary for readability and reproducibility of this research.

**No line numbers are added to the text making it hard to pinpoint review.

Abstract:

No need to mention SPSS in abstract.

Re-check keywords; use non-generic ones

Introduction:

Intro page 3, 2nd paragraph: need citation for “Many communities are unaware…”

Intro page 3, paragraphs 4 and 5 do not add anything to the introduction narrative, especially where the aim is already spelled out in 3rd paragraph

Materials and Methods

Need citation for census population numbers in “Study cite” section

Page 4:

Why 400 households in study design when the sample size calculation was 422 households?

What was the qualification requirements for the key informant components? Which personnel were selected and which were rejected? Why separate inclusion and exclusion criteria from study design?

Page 5:

Format of equations is very confusing. Use math equation editor to add equations or some other consistent method

Why 22 key informants?

Page 6:

How was the 422 household sample distributed amongst all the regions? A breakdown of each sampling design is necessary here with a table/ figure, etc.

What socio-demographic factors were recorded? Only Table 1 in results show some of the factors? Were there others? What was the survey questionnaire? Without the actual questionnaire it is hard to discern the type of quantitative data collected. How were these socio-demographic factors chosen? Were they aligned to some prior study or authors’ expertise?

Statistical significance criterion is very confusing. Do you mean p < 0.05 or p < 0.001?

What about the qualitative data collection? Were the interviews structured? Semi-structured? How was the thematic analysis performed? Which dimensions/ themes were pre-supposed?

Results and findings:

Page 7:

How were household heads defined? Were they just the respondents who were available at the time of survey or were they identified from census data? No mention of this in materials and methods.

How was tertiary education defined?

Not necessary to summarize Table 1 in text. For me, the socio-demographic characteristics text has the same information as table 1 making it redundant.

Page 8:

Was there age bias in how the questions were answered? For example, in Table 1, only 3 respondents <25 years old answered yes to Adherence to immunization? What was the criterion for adherence? Does it mean annual vaccination as defined by WHO guidelines? No mention of this in materials and methods or an appendix of questionnaire.

Page 10:

First paragraph belongs to materials and methods. Also, citations needed.

Again, the text only states what the Table 1 shows. Not necessary to spell out in text.

Page 11:

Repeat of some of the text from page 10.

2nd paragraph needs to be in materials and methods. Some discussion points are mixed in as well. Please restructure the results section carefully.

Page 12 and 13:

Were the odds ratios exponentiated? Or on log scale? What is their interpretation? For example, does B of -0.067 mean that the household breadwinner when not being a respondent is less likely to adhere to vaccination protocol? What do the numbers mean exactly? What were the reference levels for each individual factor? Was there no pairwise comparison post-hoc? What is constant in Table 2? Is it an intercept? How come only 1 degree of freedom for factors such as age with 5 levels or marital status with 5 levels, etc,.? Consider redoing the analysis properly.

How do you define positive relationship between different levels of monthly income?

Page 14 and 15:

This is again descriptive statistics. What were the exact questions? No mention in materials and methods

Page 16:

What are KII? Again were they structured? Semi-structured? How many interviewers?

Terms like Key informants and officers used inter-changeably. Are they the same or different? Consistency is lacking

Discussion:

Page 17, 2nd Paragraph: Why compare to studies from Nepal and China only? No literature for Kenya and neighboring countries on rabies awareness?

Page 17, 3rd Paragraph: Is this due to reporting age bias? Were the respondents equally aware of the questioning methods across ages?

Page 18:

What about social desirability bias? Did the respondents report what they thought the interviewers wanted to hear? Add discussion on how steps were taken to reduce that.

Page 19 and 20:

Were the factors that restricted visit to primary health clinic taken into account? What about distance for closest facility? Social factors like home remedies, were there any reported instances?

Conclusion: Too long. Summarize in 2-3 sentences. Or integrate in the discussion.

Minor points:

Abstract:

Space missing after between numbers and parentheses for percentages in several spaces

What is 70[1].6%?

Introduction:

Intro page 2, 1st Paragraph: Period missing

Materials and Methods:

Why different formatting and the style of citations compared to the rest of main body?

Page 4: Fisher et al. in Sample size

Page 5: Extra equals sign. Use math equation editor.

Results:

Why different formatting and the style of citations compared to the materials and methods and introduction?

Conclusion: Too long. Summarize.

Reference/ bibliography formatting is not consistent.

---

## [Decision Letter · Decision Letter 1]

6 Feb 2025

PGPH-D-24-02047R1

Factors Influencing Vaccination of Dogs against Rabies within Household Heads in South Mugirango Sub-County-Kisii County; Kenya

Dear Dr. Fanice Kerubo Obara,

Thank you for submitting your manuscript to PLOS Global Public Health. After careful consideration, we feel that it has merit but does not fully meet PLOS Global Public Health’s publication criteria as it currently stands. Therefore, we invite you to submit a revised version of the manuscript that addresses the points raised during the review process.

Additional comments:

Reviewer #1: The abstract currently presents the study's content, but it could benefit from a more explicit emphasis on the central aim of the research. Since the study's primary objective is to assess factors influencing dog vaccination against rabies among household heads, I suggest highlighting this key focus more prominently. Additionally, while the abstract provides an overview, refining the language will help align it with academic standards and enhance clarity. A careful review by a native English speaker would further improve the readability and ensure the scientific tone is appropriately maintained throughout.

Reviewer #2: General Comments: I commend the authors for making comprehensive and detailed changes to the manuscript. I have the following comments:

All sections of the main body:

Check for grammar and syntactical errors like missing commas, periods. There are multiple missing spaces between sentences. The manuscript should have consistent writing style and formatting, please check with a language editor if deemed necessary.

No data availability statement was found in the manuscript.

Abstract: No comments

Introduction:

Last three paragraphs describe the impacts and the aims of the study. Try to combine last three paragraphs or shorten them.

Materials and Methods;

“Participants for the quantitative survey were recruited based on the inclusion criteria of being household heads who owned at least one dog and provided informed consent.” repeated twice; once in recruitment of participants and once in inclusion criteria. Remove one.

“Statistical significance was set at p<0.05p < 0.05p<0.05.” Repeated terms?

“Qualitative data were analyzed thematically, with data grouped into emerging themes.” Explain how? Given the reference to a previous study, it would be better to summarise it in a sentence or two.

“In South Mugirango Sub-County, several Socio-demographic factors play a role in shaping the immunization practices of Household Heads (HHs) concerning their dogs.” Is this author’s expertise?

Results:

Paragraphs between: “For gender, the data revealed…. Respondents with no formal education had negligible representation, with 0% adhering.” - Redundant. Only reiterating what is in the table. Instead add interpretation of the findings in Table 1.

“The logistic regression analysis aimed to identify Socio-demographic factors that influence adherence to dog vaccination in South Mugirango.” How? What is the dependent variable. What do the coefficients mean? Are they odds ratios for vaccination? Please explain the use and application of Logistic regression to help readers understand what this analysis is about.

Table 2: Are the coefficients and CI on log scale or natural scale? Seems to me that the coefficients have not been exponentiated while the CI is exponentiated. Why? No mention in footnotes.

“The interviews with livestock and public health officers corroborated the quantitative findings, emphasizing the significant role of socio-demographic factors in vaccination adherence.” Where are the specific results of the interview pertaining to this?

“The assessment of knowledge and practices related to rabies vaccination, based on key informant interviews (KIIs), revealed several important insights:” Where is the thematic analyses mentioned in Materials and Methods? How were the themes mentioned in this subsection generated? How many key informant respondents responded or did not respond to particular theme? What was the nature of the semi-structured interview that demarcated these themes?

Discussion:

A lot of information in the discussion, especially pertaining to the indicators of socio-demographic characteristics needs to be mentioned in the Introduction section at least. Refer to my previous comment “ Is this author’s expertise?” The information in the discussion addresses this comment and comes too late out of the blue in the text. Major restructuring to the introduction with respect to the discussion section is necessary here to make both sections coherent.

**Note to editor: No line numbers makes it very difficult for me to write precise comments.

We look forward to receiving your revised manuscript.

Kind regards,

Sukanta Chowdury, Ph.D

Academic Editor

---

## [Editor Report · Decision Letter 2]

20 Mar 2025

PGPH-D-24-02047R2

Factors Influencing Vaccination of Dogs against Rabies within Household Heads in South Mugirango Sub-County-Kisii County; Kenya

Dear Dr. Fanice Kerubo Obara,

Thank you for submitting your manuscript to PLOS Global Public Health. After careful consideration, we feel that it has merit but does not fully meet PLOS Global Public Health’s publication criteria as it currently stands. Therefore, we invite you to submit a revised version of the manuscript that addresses the points raised during the review process.

We look forward to receiving your revised manuscript.

Kind regards,

Sukanta Chowdury, Ph.D

Academic Editor

Additional Editor Comments:

Abstract:

- You included more texts in the introduction section and less texts in the result section

- Suggest to include more information in the result section and reduce words from introduction section

Results:

- Suggest to include study period

- How many key informant interviews conducted? Mention

- I did not find revised table 1. It is deleted. Why

Every study has limitations. Please add a paragraph for limitation after discussion section.

Conclusions looks so long. First paragraph is fine. Suggest to relocate last three paragraphs in the discussion section.

---

## [Editor Report · Decision Letter 3]

28 Mar 2025

Factors Influencing Vaccination of Dogs against Rabies within Household Heads in South Mugirango Sub-County-Kisii County; Kenya

PGPH-D-24-02047R3

Dear Dr. Fanice Kerubo Obara,

We are pleased to inform you that your manuscript 'Factors Influencing Vaccination of Dogs against Rabies within Household Heads in South Mugirango Sub-County-Kisii County; Kenya' has been provisionally accepted for publication in PLOS Global Public Health.

Best regards,

Sukanta Chowdury, Ph.D

Academic Editor
